# Unveiling contextual realities by microscopically entangling a neutron

J. Shen [1,2,3,9], S.J. Kuhn[1,2,3,9], R.M. Dalgliesh [4], V.O. de Haan[5], N. Geerits[6], A.A.M. Irfan[1,3], F. Li[7], S. Lu [1,3], S.R. Parnell[8], J. Plomp[8], A.A. van Well[8], A. Washington [4], D.V. Baxter[1,2,3], G. Ortiz[1,3], W.M. Snow[1,2,3] & R. Pynn [1,2,7 ✉]

The development of qualitatively new measurement capabilities is often a prerequisite for critical scientific and technological advances. Here we introduce an unconventional quantum probe, an entangled neutron beam, where individual neutrons can be entangled in spin, trajectory and energy. The spatial separation of trajectories from nanometers to microns and energy differences from peV to neV will enable investigations of microscopic magnetic correlations in systems with strongly entangled phases, such as those believed to emerge in unconventional superconductors. We develop an interferometer to prove entanglement of these distinguishable properties of the neutron beam by observing clear violations of both Clauser-Horne-Shimony-Holt and Mermin contextuality inequalities in the same experimental setup. Our work opens a pathway to a future of entangled neutron scattering in matter.

[1] Department of Physics, Indiana University, Bloomington, IN 47405, USA. [2] Indiana University Center for the Exploration of Energy and Matter, Bloomington, IN 47408, USA. [3] Indiana University Quantum Science and Engineering Center, Bloomington, IN 47408, USA. [4] ISIS Pulsed Neutron and Muon Source, Rutherford Appleton Laboratory, Chilton, Oxon OX11 0QX, UK. [5] BonPhysics Research and Investigations BV, Laan van Heemstede 38, 3297AJ Puttershoek, The Netherlands. [6] Atominstitut, TU Wien, Stadionallee 2, 1020 Vienna, Austria. [7] Neutron Sciences Directorate, Oak Ridge National Laboratory, Oak Ridge, TN 37830, USA. [8] Faculty of Applied Sciences, Delft University of Technology, Mekelweg 15, 2629 JB Delft, The Netherlands. [9] These authors contributed equally: J. Shen, S. J. Kuhn. ✉email: rpynn@indiana.edu

Amost amazing aspect of quantum reality is the possibility to share information non-locally between two or more space-like separated subsystems, a spooky action at a distance, as Einstein liked to call it and Bell epitomized in an inequality[1,2]. The fact that measuring sets of compatible observables does not unveil predetermined physical properties, as pointed out by Kochen and Specker[3,4], reveals the contextual nature of quantum measurements. Behind all these non-classical statistical correlations is the property of entanglement wherein the state of the whole is more than the sum of its (constituent) parts[5]. Developing quantum probes that exploit these correlations as a means for investigating entanglement in matter could lead to insights into some of the most interesting materials studied today, such as frustrated magnets hosting quantum spin liquids and unconventional superconductors with strange metallic behavior[6].

In this paper we propose and demonstrate a quantum probe consisting of a tunable beam of entangled neutrons that promises transformative scientific and technological advances. Neutron scattering is a well-established technique to probe the structure and dynamics of materials. Neutrons can penetrate deep into most materials while the independent states of neutron energy, trajectory, and spin polarization can be coherently controlled. However, using a quantum entangled neutron beam to perform neutron scattering has not been attempted. A composite pure state of a single neutron is defined as entangled if it cannot be determined by the states of its constituent subsystems, in this case its spin, trajectory, and energy, each of which is associated with a distinguishable property of the single-particle neutron[7,8]. Our distinguishable-subsystem structure is definitely associated with degrees of freedom other than the neutrons themselves; in other words, our neutron beam is mode-entangled as opposed to particle-entangled[7].

The trajectory of a particular neutron spin state can be manipulated using refraction through magnetic fields that couple with the neutron's magnetic moment, which is tied to its spin. Our experiments use a neutron spin-echo interferometer to show that one can entangle the path, spin and energy of individual neutrons on a microscopic length scale. The spatial separation of neutron-spin paths, which we refer to as the entanglement length, is adjustable from tens of nanometers to microns, limited only by neutron beam divergence and magnetic field aberrations. Energy separations can be varied from peV to neV for neutron energies of a few meV, constrained by the frequency range of RF neutron spin flippers. Prior work conducted using single-crystal neutron interferometers has validated the predictions of quantum mechanics for two-subsystem or three-subsystem neutron entanglement[9]. Such devices have also been used to demonstrate the realization of a decoherence-free subspace in neutron interferometry[10] and neutron mixed orbital angular momenta[11,12]. We present a much more flexible instrument that can carry out entangled neutron scattering experiments on material samples over dynamically-relevant scales in space and energy. We exploit this flexibility to demonstrate independent control of distinct quantum subsystems and to generate and characterize a variety of entangled neutron states.

In this article, we prove that the neutron beam is entangled by designing an interferometer to measure neutron spin-polarizations, organized in terms of contextuality witnesses, which violate the relevant contextuality inequalities. We can tune the number of entangled subsystems from two (spin-path) to three (spin-path-energy) to test for violation of Clauser-Horne-Shimony-Holt (CHSH) or Mermin contextuality inequalities, respectively. The CHSH contextuality inequality employs commutable spin and path projection operators to define an expectation value for a Bell state at specific spin and path values[13,14]. Comparison of these expectation values is used to confirm the contextual nature

of reality. The Mermin contextuality inequality was introduced in a neutron interferometry experiment in order to validate the idea of entanglement in a system with more than two distinguishable subsystems[15]. Here we present a CHSH value of $S = 2.16 \pm 0.01 + 0.02$ and a Mermin witness value of $M = 3.052 \pm 0.007 + 0.017$, where the ± error bars are the standard deviations of random errors and the final error is an upper bound on systematic effects. Both of these witnesses, whose detailed definitions are provided in Eqs. (1) and (3) below, are well above the non-contextual bounds and are very close to the maximum possible value for the neutron beam polarization used. These two results are realized in the same experimental setup.

## Results

**Experiment**. The experiment was conducted using the Larmor instrument at the ISIS pulsed neutron source in the UK. Larmor has four radio-frequency (RF) neutron spin flippers, each with a static magnetic field along the vertical z direction, through which the neutron beam travels in the y direction as shown in Fig. 1a. The instrument acts as an interferometer in which the first two RF flippers entangle the neutron spin, path, and (optionally) energy subsystems whereas the third and fourth RF flippers act as a disentangler[16]. Before the RF flippers, a magnetic supermirror polarizes the neutron spin state in the z direction, producing a beam that is predominantly spin up $|\uparrow\rangle$. The neutron spin then evolves adiabatically to the x direction guided by a gradually changing magnetic field and encounters a $\pi/2$ flipper which non-adiabatically separates the x-directed magnetic field before the $\pi/2$ flipper and a z-directed field after the $\pi/2$ flipper. This device produces a quantum superposition $\frac{1}{\sqrt{2}}(|\uparrow\rangle + |\downarrow\rangle)$ and triggers the start of the spin precession, i.e. a time-dependent evolution of the relative phase of the up and down states. A small, z-directed guide field is applied between the first $\pi/2$ flipper and the $\pi$ flipper in the middle of the instrument. This $\pi$ flipper is a current sheet that sharply separates a magnetic field along z before the flipper and a field along −z, which is applied between the $\pi$ flipper and the final $\pi/2$ flipper.

Up and down spin states have opposite Zeeman energies from the static field in the RF flipper region and hence different kinetic energies and velocities by energy conservation. Since the boundaries of the static flipper field are angled to the neutron beam, the opposite spin states are refracted in different directions at these boundaries, creating a two-path-state subsystem indicated in Fig. 1a, which shows the evolution of the paths (green lines) of the up and down spin states (blue arrows) through each of the RF flippers. A Bell state which contains the spin subsystem and the path subsystem of $|1\rangle$ and $|2\rangle$ is thus prepared after the second RF flipper. The separation of the neutron-spin paths, the entanglement length, is given by $\xi = c\lambda_{n}^{2}$, where $c$ is 9770 nm$^{-1}$ for our experimental configuration and $\lambda_{n}$ is the neutron wavelength. In our experiment the entanglement length was $\xi \sim 1.6$ microns for a neutron wavelength of 0.4 nm [see Methods].

We manipulate the phase difference between the neutron spin states (spin phase, denoted by $\alpha$) using a variable, z-directed, static, magnetic field indicated schematically in Fig. 1a. The analogously defined path phase, $\chi$, is adjusted by passing the beam through single-crystal, rectangular-parallelepiped, quartz blocks that make equal and opposite angles to the average neutron beam direction, as shown in Fig. 1a insert. This arrangement makes the same relative path phase as the isosceles quartz triangle shown in Fig. 1a and presents different distances through the quartz for the two paths [see Methods]. The block angles, labeled $\phi$ in Fig. 1a, can be set reproducibly to one of ten preselected values, allowing the path phase to be varied over a range of approximately $-3\pi/2$

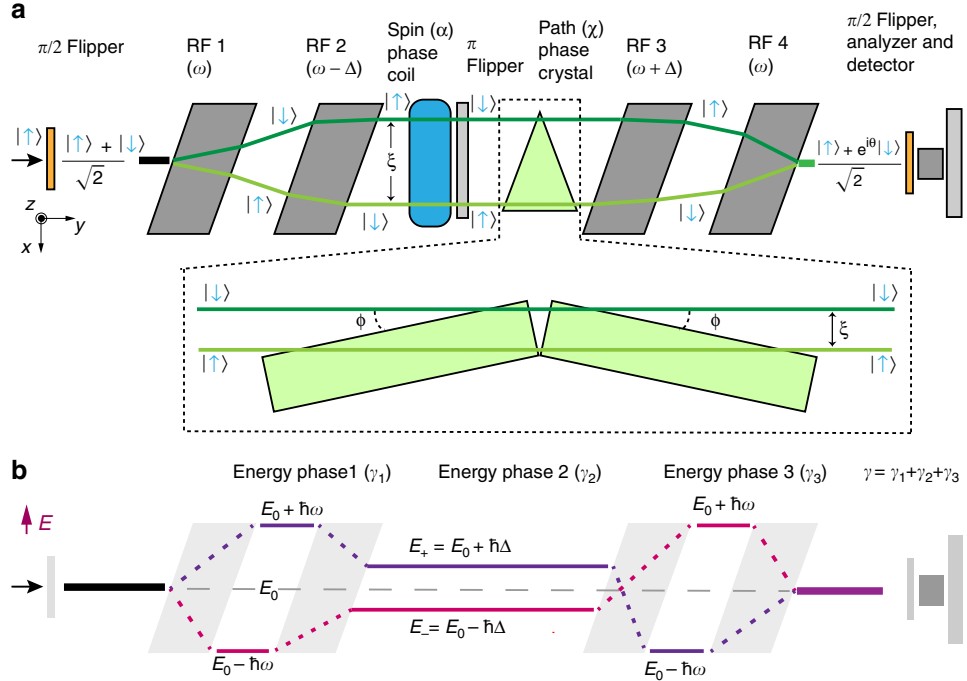

**Fig. 1 Sketch of the paths and energies of neutron spin states in the interferometer. a** Schematic plan view of the main spin manipulation components of the Larmor instrument showing the evolution of the neutron path and spin states along the beam line with neutrons propagating from left to right in the y direction. A small z-directed field is applied between the first $\pi/2$ flipper and the $\pi$ flipper in the middle of the instrument. After the $\pi$ flipper, the applied field is reversed. A superposition of up and down spin states, produced by a $\pi/2$ flipper at the beginning of the instrument passes through two RF flippers (RF1 and RF2) whose static magnetic fields cause the up and down states to be refracted along different paths that are separated by the entanglement length, $\xi$, in the space between the second and third RF flippers. In the absence of the spin-phase coil (blue) and path-phase crystal (green), the two states are disentangled by the second pair of RF flippers (RF3 and RF4) to yield a spin echo whose polarization is measured using a $\pi/2$ flipper, a polarization analyzer and a neutron detector. The phase change, $\theta$, is defined as the sum of the subsystems phases. A small additional z-directed magnetic field (blue) is used to alter the relative phases of the two spin states. Two rectangular quartz crystals, shown in more detail within the dashed-line box, cause the two neutron paths to accumulate different phases as the paths travel different distances in air and in quartz. **b** A plot of the total neutron energy for each neutron spin state along the beam line. Each RF flipper reverses a neutron spin state at the same time as it exchanges a quantum of RF energy with that state. A difference in the energy phase between the two spin states develops in the space between each pair of RF flippers because the two states have different total energies. In the normal spin-echo condition, $\Delta = 0$ and the energy phase developed between the first two RF flippers ($\gamma_1$) is canceled by that developed between the second pair of RF flippers ($\gamma_3$). When $\Delta \neq 0$ the total energy phase ($\gamma$) is non-zero.

to $3\pi/2$ radians for 0.4-nm-wavelength neutrons. After the spin and path phases have been set, the neutron spin is reversed by a $\pi$ flipper before traveling through the third and fourth RF flippers which disentangle the states. A final $\pi/2$ flipper projects the neutron spin onto the x axis where the neutron intensity is measured using a supermirror spin analyzer and a set of $^3$He tube detectors.

The total neutron energy is altered by each of the RF flippers, as shown in Fig. 1b, and may be entangled after the second RF flipper in a Greenberger-Horne-Zeilinger (GHZ) state with the path and spin subsystems by varying the frequencies of the RF flippers. The RF flippers work by exchanging a quantum of RF energy with the neutron, changing the total energy of the neutron at the same time as they flip the spin state from up to down or vice versa. There are three well-defined total energy phases, one between each adjacent pair of RF flippers, which sum to a total energy phase $\gamma = \gamma_1 + \gamma_2 + \gamma_3$ [see Methods]. When the four RF flippers are set to the same frequency $\omega$, $\gamma_1 = -\gamma_3$ and $\gamma_2 = 0$ leaving the energy state unentangled, so the neutron is in the spin and path entangled Bell state $|\Psi_{\text{Bell}}\rangle = \frac{1}{\sqrt{2}}(|\uparrow 1\rangle + |\downarrow 2\rangle)$ after the second RF flipper. To entangle the energy, RF flippers 2 and 3 are set to frequencies of $\omega - \Delta$ and $\omega + \Delta$ respectively so that $\gamma_2$ is non-zero, leading to a net difference in the energy phase between the spin states. The energy subsystem is denoted by $|E_+\rangle$ and $|E_-\rangle$, where $E_\pm = E_0 \pm \hbar\Delta$, $E_0 = h^2/(2m_n\lambda_n^2)$, and $m_n$ is the

neutron mass. This gives a triply entangled GHZ state $|\Psi_{\text{GHZ}}\rangle = \frac{1}{\sqrt{2}}(|\uparrow 1E_-\rangle + |\downarrow 2E_+\rangle)$[17]. The contextual realities for both the Bell state and the GHZ state can be tested with the Larmor instrument.

**Entangling spin and path**. The first series of measurements were performed on the two-subsystem entangled Bell state ($\Delta = 0$) so that only the neutron spin and path were entangled. The beam intensity was measured for a range of spin and path phases, as shown by the black circles in Fig. 2a. The values of the spin and path phases were determined by fitting the measured neutron polarization $P$ as a function of neutron wavelength ($\lambda_n$)[18] to the function $P(\lambda_n) = P_0(\lambda_n)\cos\left([a - a_0]\lambda_n + b\lambda_n^3 + \varphi_0\right)$ where $P_0(\lambda_n)$ is the beam polarization with $\alpha = \chi = 0$ and $a_0$ as well as $\varphi_0$ are constant correction terms (see Methods). $a\lambda_n$ and $b\lambda_n^3$ yield $\alpha$ and $\chi$ respectively at any neutron wavelength. The intensity dependence on the spin and path phases is shown in Fig. 2b. The intensity/spin/path dataset is fit by the function: $N = A \times \cos(\alpha + \chi) + B$, where $N$ is the neutron intensity, and $A$ and $B$ are the amplitude and background respectively. Figure 2c shows this function fits the data well.

The contextuality of the measurement for two neutron distinguishable subsystems, path and spin, is evaluated using a CHSH inequality which sets a limit for the non-contextual

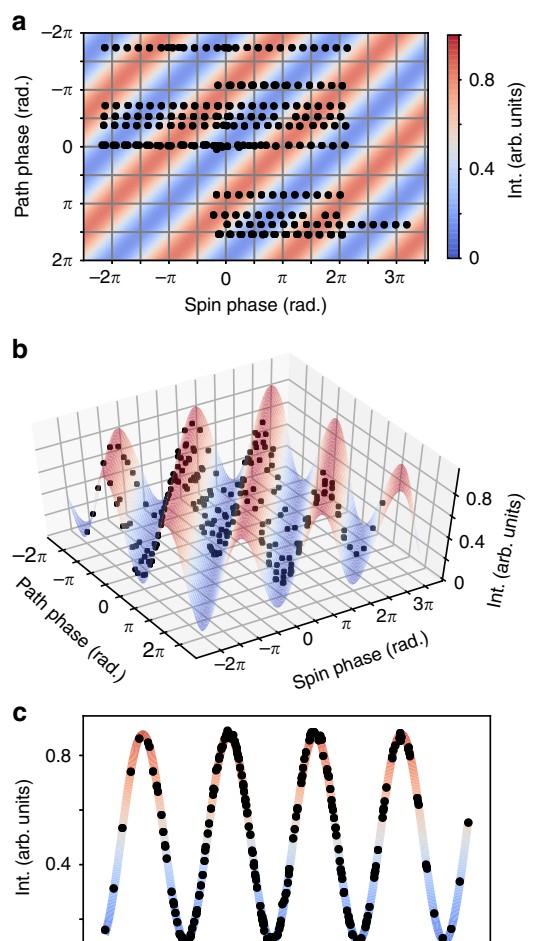

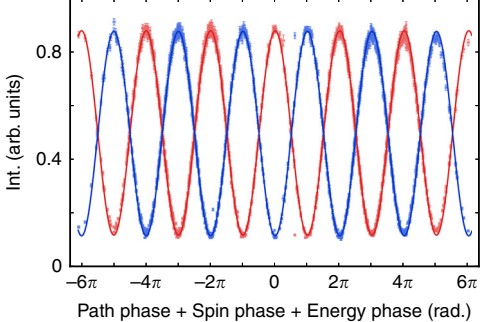

**Fig. 3 The complete dataset and fitting curve for the triply entangled state.** The spin, path, and energy phases are summed and plotted vs. the normalized intensity. The red (blue) data points correspond to neutrons in the up (down) state before the first $\pi/2$ flipper. The fit contrast for the red curve is 0.768 and for the blue curve is 0.763.

values are computed directly from the values of the cosine fit at the designated spin and path positions and the $\pi$ shifted positions, i.e.:

$$E(\alpha,\chi) = \frac{N(\alpha,\chi) - N(\alpha,\chi+\pi) - N(\alpha+\pi,\chi) + N(\alpha+\pi,\chi+\pi)}{N(\alpha,\chi) + N(\alpha,\chi+\pi) + N(\alpha+\pi,\chi) + N(\alpha+\pi,\chi+\pi)}$$

(2)

Using the fitted values of $A = 0.379(0.001)$ and $B = 0.49(0.002)$ to evaluate the intensities at the above position gives expectation values of $S = |0.541(0.006) + 0.541(0.006) + 0.541(0.006) + 0.541 (0.006)| = 2.16 \pm 0.01 + 0.02$, well above the classical limit of 2, with a statistical error of $\pm0.01$ and a bound on the systematic error of $+0.02$. In this experiment as well as in the single-crystal interferometry experiment, the primary factor decreasing the inequality value is the neutron polarization[19]. The peak polarization for our experiment was 0.78 which is achieved at a wavelength of 0.4-nm. This sets the maximum value of $S$ that we could see on the Larmor beamline for the CHSH inequality to $2\sqrt{2} \times 0.78 = 2.20$. With this polarization, the expected maximum value for a classical system should be $2 \times 0.78 = 1.56$. This measurement demonstrates the ability to entangle neutrons on a microscopic length-scale with a conventional neutron scattering instrument.

**Entangling spin, path and total energy.** For the second experiment we entangle the energy state with the neutron spin and path states to obtain a Greenberger-Horne-Zeilinger (GHZ) state. The neutron spin and path states behave in the same way as in the $\Delta = 0$ case, while the energy phase value can be calculated directly from the frequency shift $\Delta$. Once again, the contextuality witness is defined in terms of the expectation values[20] extracted from a cosine fit to the neutron count rates given by $N = A \times \cos(\alpha + \chi + \gamma) + B$, where $\gamma$ is the energy phase. This data closely follows the predicted sinusoidal shape as shown in Fig. 3. The Mermin witness relates to the triply entangled expectation values by:

$$M = E[\sigma_x^s \sigma_x^p \sigma_x^e] - E[\sigma_x^s \sigma_y^p \sigma_y^e] - E[\sigma_y^s \sigma_x^p \sigma_y^e] - E[\sigma_y^s \sigma_y^p \sigma_x^e]$$

(3)

Here $\sigma_{x,y}^e$ refers to the Pauli matrix of the two-level subsystem for the energy. In this case, $x$ and $y$ correspond to the 0 and $\frac{\pi}{2}$ angles for $\alpha$, $\chi$, and $\gamma$. Expectation values are calculated from the 3D fitted intensities in a manner similar to the doubly entangled system. To write the expression more cleanly we use the

**Fig. 2 Measured path and spin phases. a** The black circles indicate the path and spin phases where measurements were taken at a neutron wavelength of 0.4 nm. The colored stripes illustrate the cosine curve fitted to the data. **b** 3D view of the normalized intensities (black circles) plotted against the path and spin phases and fitted by a cosine fitting function. **c** Measured intensities and the fitted cosine function plotted against the sum of path and spin phase. Error bars, representing standard deviations derived from counting statistics, are the size of the markers or smaller.

measurement of expectation values. A classical system has a strict upper limit of 2, while the expected value for a contextual measurement is $2\sqrt{2}$. This manifests itself in the predicted sinusoidal behavior of the neutron intensity. This inequality, which has previously been used to demonstrate quantum contextuality in a single-crystal neutron interferometer[19], is evaluated by measuring expectation values, $E(a,\chi) = E[\sigma_u^s \sigma_v^p] = \langle\Psi_{Bell}|\sigma_u^s \sigma_v^p|\Psi_{Bell}\rangle$, where $\sigma_u^s = \cos(\alpha)\sigma_x^s + \sin(\alpha)\sigma_y^s$ and $\sigma_v^p = \cos(\chi)\sigma_x^p + \sin(\chi)\sigma_y^p$. Here, $\sigma_{x,y}^{s,p}$ refer to the $x$ and $y$ Pauli matrices of the two-level subsystems for the spin and path and $u$ and $v$ refer to the projection axes rotated from the x axis by $\alpha$ and $\chi$, respectively. We expect the maximum violation of the CHSH inequality when $\alpha_1 + \chi_1 = -\pi/4$ and that $\alpha_2 - \alpha_1 = \chi_2 - \chi_1 = \pi/2$. The contextual witness is defined as:

$$S = |E(\alpha_1, \chi_1) + E(\alpha_1, \chi_2) + E(\alpha_2, \chi_1) - E(\alpha_2, \chi_2)|$$

(1)

We have chosen $\alpha_1 = 0$, $\alpha_2 = \pi/2$, $\chi_1 = -\pi/4$, and $\chi_2 = \pi/4$ for a neutron wavelength of 0.4 nm where the neutron beam polarization and statistical accuracy are both optimized. The expectation

definition $N_{\mu_s\mu_p\mu_e} = N(\alpha + \mu_s\pi, \chi + \mu_p\pi, \gamma + \mu_e\pi)$ and

$$E\left[\sigma_{x,y}^s \sigma_{x,y}^p \sigma_{x,y}^e\right] = \sum_{\mu_s,\mu_p,\mu_e=0,1} (-1)^{\mu_s+\mu_p+\mu_e} N_{\mu_s\mu_p\mu_e} / \sum_{\mu_s,\mu_p,\mu_e=0,1} N_{\mu_s\mu_p\mu_e}$$

(4)

With these parameters we find a Mermin witness value of $M = 3.052 \pm 0.007 + 0.017$, where we have again included both the statistical error of $\pm 0.007$ and a systematic error bound of $+0.017$. The value of $M$ is well above both the classical value of 2 and the $2\sqrt{2}$ limit for a two-subsystem entangled neutron, confirming that all three independent states are entangled. The expected $M$ value for a contextual measurement with the neutron beam polarization we used is $0.78 \times 4 = 3.12$, close to the value from our measurement.

**Pulsed versus continuous neutron sources.** Although these measurements were made at a pulsed neutron source using time-of-flight methods, they could equally well have used a continuous neutron source. The 2.5% wavelength band and milliradian beam divergence that we used are easily achievable at a continuous neutron source using conventional methods. At a continuous source the total neutron intensity might be larger than at a pulsed source. On the other hand, time-of-flight methods allow for an easy determination of the path and energy + spin phases for each configuration. At a continuous source, these phases would need to be calibrated before the experiment and checked during the measurement to ensure that drifts did not occur. The neutron intensity and wavelength range for neutron entangled-state experiments are expected to be similar to those for conventional polarized neutron measurements.

## Discussion

In summary, we have developed a quantum probe comprising a tunable beam of entangled neutrons, with the potential to reveal quantum correlations in microscopically entangled matter. In this work we focused on controlling and proving the entanglement characteristics of the neutron probe. We showed that the distinguishable properties of spin, path and energy of individual neutrons can be entangled in a controllable fashion. Contextual realities have been unveiled by carefully crafted interferometers that prepare maximally entangled states and measure appropriate witnesses that test violation of contextuality inequalities. The entangled neutron probe also offers opportunities to investigate fundamental symmetries and interactions such as the effect of gravitation on the entangled distinguishable properties of the neutron, such as spin. Neutron orbital angular momentum can be added to the list of additional properties to be entangled, thus advancing applications in quantum metrology and high-precision measurement. With these results, we have demonstrated the availability of a tool that can be used to push the limits of quantum-enhanced metrology in exploring the frontiers of fundamental physical phenomena[21,22]. All these exciting themes will be subjects for future experiments.

## Methods

**Experiment details.** The Larmor instrument, located at the second target station of the ISIS neutron and muon source, part of Rutherford Appleton Laboratory (RAL) in the UK, was used to perform this experiment. The ISIS second target station is a 10 Hz pulsed neutron source with event-mode data acquisition so that the time-of-flight (TOF), and hence the wavelength, of each detected neutron is recorded. A low-efficiency neutron monitor detector in the incident neutron beam is used for intensity normalization. The neutron beam size used during the experiment was set by two rectangular apertures separated by ~4.83 m. The first aperture, located before the first $\pi/2$ flipper measured 5 mm × 5 mm while the second, measuring 10 mm high and 3 mm wide was located immediately before the quartz blocks used to manipulate the neutron path phase. The transverse coherence length of neutron wave-packets is much smaller than the entanglement length at all points along the neutron beam line. Neutrons are polarized by magnetic supermirrors and an

optional $\pi$−flipper allows either up or down neutron polarization states to be used in the experiment. Four RF flippers operated in resonant mode are placed on the beamline in two sets of two (cf Fig. 1a). Each pair is separated by 1.2 m. The RF flippers are tuned with a vertical (z-direction) static magnetic field of $B_0 = 34.29$ mT to match the resonance condition at 1 MHz. The strength of the RF magnetic field is varied with time to efficiently flip neutrons' spin for wavelengths between ~0.3 nm and ~1 nm. As shown in Fig. 1a, the static field of the RF flippers ($B_0$) has borders that makes an angle of 30° with the neutron beam. The entanglement length was calibrated using the spin echo signal obtained with a 2-micron-period silicon grating and found to be $\xi = c\lambda_n^2$ with $c = 9770 \pm 80$ nm$^{-1}$.

The neutron spin direction was controlled along the beamline. A ~ 1 mT guide field in the z-direction was maintained between the first $\pi/2$ flipper and the $\pi$ flipper. A sharp change of guide field to the minus-z direction occurs at the $\pi$ flipper and this guide field is maintained until the final $\pi/2$ flipper. Each $\pi/2$ flipper provides a sharp transition between x-directed and z-directed magnetic fields, allowing a superposition state to be generated by the first $\pi/2$ flipper and the x-component of the neutron polarization to be projected by the second $\pi/2$ flipper. The Swiss Neutronics supermirror-bender polarization analyzer was made of sheets of FeCoV/TiN aligned horizontally to avoid any spatial change of analyzing efficiency in the horizontal (x-y) plane since the quartz blocks, used to control the path phase, refract the neutron beam in this plane.

The neutron energy phase accumulated between two RF flippers is given by $\gamma_i = E_i T_i / h$, where $E_i$ is the total energy of the neutron just after the first of the two flippers and $T_i$ is the time for neutrons of a particular wavelength to travel between the two RF flippers. The apparatus is initially set up in spin-echo mode with all RF flippers set to a 1 MHz frequency so that the energy phase accumulated between the first two RF flippers ($\gamma_1$) is equal and opposite to that accumulated between the third and fourth RF flippers ($\gamma_3$) for any wavelength.

Neutron intensity was measured using eight, parallel, 8 mm-diameter, 15-bar, $^3$He detector tubes aligned along x to maintain a uniform detector efficiency in this direction. The detection efficiency is estimated to be 91% at a neutron wavelength of 0.4 nm. For the analysis of our data, neutrons between 0.395 and 0.405 nm were binned to form the intensity for the CHSH and GHZ measurements. Background in the detectors is small and had no significant effect on the data analysis. Each combination of spin and path phase was measured both with spin-up and spin-down neutrons for six minutes. When the energy was also entangled, each phase combination was measured for 3 min for each neutron spin state. A total of 1376 combinations of the various phases were measured. All neutron intensities were divided by the monitor count and statistical errors were propagated through the data analysis.

Four rectangular, single-crystal quartz blocks with optically smooth faces were purchased from VM-TIM. The blocks measured 100 mm by 10 mm by 50 mm. Quartz was chosen because it has a relatively large coherent scattering cross section for neutrons and hence produces a path phase that is significantly different from the path phase accumulated in air. Additionally, quartz has a small incoherent scattering cross section, minimizing background, as well as a very small absorption cross section. The beam attenuation was measured for each configuration of the blocks and the intensity data were corrected accordingly. The orientations of the quartz blocks in the neutron beam were set reproducibly by placing them against the sides of fixed, triangular, aluminum plates whose angles were chosen to give a spread of path phases that covered a range between $-3\pi/2$ and $3\pi/2$ for 0.4-nm-wavelength neutrons. For each measurement, neutrons passed through two blocks mounted in a V configuration as shown in the second part of Fig. 1a. The path phase is given by $\chi = 2 \lambda_n \xi \rho (\cot(\phi) + \cot(\pi/2 - \phi))$, where $\rho$ is the scattering length density of quartz. This configuration was chosen to eliminate dependence of the path phase on the divergence angle of a neutron trajectory.

**Data treatment.** For each combination of spin, path and energy phases, the neutron polarization was measured as a function of neutron time-of-flight and normalized using the polarization with $\alpha = \chi = \gamma = 0$. The normalized polarization is well fitted over a wavelength range 0.38–0.8 nm by the quotient of two cosine functions $\cos([a - a_0]\lambda_n + b\lambda_n^3 + \varphi_0) / \cos(a_0\lambda_n - \varphi_0)$ where $a_0$ and $\phi_0$ are constants that account for small errors in tuning the spin echo and for slight missetting of the phases of the RF flippers, respectively (see Supplementary Fig. 1). The relative path phase between the two neutron spin states varies as $\lambda_n^3$ while the relative spin and energy phases vary as $\lambda_n$. The energy phase was calibrated by comparing polarizations obtained with different RF frequency shifts, $\Delta$, with a known spin phase. The energy phase for each frequency was found to closely match that expected from the value of the frequency shift, and the value calculated from the (very stable) RF frequency was used for the energy phase in all further data analysis. The fitted path phases were found to match those calculated from the instrument parameters to within $0.02\pi$ radians.

The statistical errors quoted for the witnesses are standard deviations that arise only from the propagation of counting statistics. A Monte Carlo method was used to calculate the final statistical error and to check its distribution and confidence level. We also include potential systematic errors that are derived by using different methods of data analysis. For example, the path phases can either be found by fitting the time-of-flight polarization data or can be deduced from the measured angles of the quartz blocks and their known neutron scattering properties. These two methods give slightly different results so we quote the witness values found by

fitting the wavelength-dependent polarization data and cite a potential systematic error resulting from the alternative analysis method.

## Data availability

Raw data from the experiments described here are available at https://data.isis.stfc.ac.uk/doi/investigation/100757753.

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

## Acknowledgements

We thank Prof. Y. Hasegawa for useful discussions. Experiments at the ISIS Neutron and Muon Source were supported by a beamtime allocation RB1820192 from the Science and Technology Facilities Council. W.M.S. acknowledges NSF PHY-1614545 and the IU Center for Spacetime Symmetries. A number of the authors acknowledge support from the US Department of Commerce through cooperative agreement number 70NANB15H259. F.L. acknowledges the Laboratory Directed Research and Development program of Oak Ridge National Laboratory. The IU Quantum Science and Engineering Center is supported by the Office of the IU Bloomington Vice Provost for Research through its Emerging Areas of Research program. The work described in this paper arose from the development of magnetic Wollaston prisms and RF flippers funded by the US Department of Energy through its STTR program (grant numbers DE-SC0009584 and DE-SC0017127).

## Author contributions

The experiment was conceived by D.V.B., G.O., R.P. and W.M.S.; critical input on experimental design was provided by J.S., R.M.D., S.R.P., J.P., A.A.v.W. and R.P.; the experiment was conducted by J.S., S.J.K., R.M.D., V.O.d.H., N.G., F.L., S.R.P., A.A.v.W., and A.W. under the leadership of R.P.; data analysis was done principally by JS with contributions by S.J.K. and V.O.d.H.; theoretical work was done by A.A.M.I. and S.L. led by G.O.; the paper was written by S.J.K., G.O. and R.P. and edited, after comments by other authors, by G.O. and R.P.

## Competing interests

The authors declare no competing interests.
