## [Peer Review File · Nature Communications]

Reviewers' Comments:

Reviewer #1:

Remarks to the Author:

I suggest accepting the manuscript with a few minor edits:

1. reduce the unnecessary use of "entanglement" throughout the article. For example in abstract the phrase "entanglement length" is used, I do not think that that is a proper term.
2. incorporate a more extensive discussion on: single particle entanglement (correlations between the degrees of freedom) vs the entanglement between different particles. Just for clarity.
- 3 The experimental setup is similar to <https://www.pnas.org/content/116/41/20328> (which also deals with entanglement between different degrees of freedom, spin and OAM) they should appropriately cite the work.

Reviewer #2:

Remarks to the Author:

From the point of view of neutron interferometry, this paper is important for several reasons. First, the work shows both the high sensitivity and high precision of neutron interferometry as performed on the ISIS LARMOR spin-echo SANS beamline.

As best the reviewer can tell, the LARMOR beamline was used without modification. The sample consisted of a phase prism with an adjustable vertex angle. This means the experiment pioneered herein is readily duplicated elsewhere, provided beamtime at the world's highly oversubscribed beamlines can be acquired. The publication of this work will certainly improve the odds for successful beamtime proposals.

The work ends with the statement of potentially widespread applications such as "gravity". This statement is not backed up with a reference. This reviewer recalls two presentations on a strategy for the use of grating-based interferometry to measure the gravitational constant with 1000-times better accuracy than hitherto possible. The grating-based method is classical; one wonders about the measurement precision with this entanglement method. In summary, the statement of highly interesting applications is strongly supported.

This will be the third neutron interferometry quantum entanglement publication in Nature: Hasegawa in 2003 and Clark in 2015. The other two publications have been extremely useful to the community. This work should have similar or stronger impact.

Les Butler
Louisiana State University

Reviewer #3:

Remarks to the Author:

The manuscript describes a novel experimental technique to entangle neutron beam in spin, trajectory and energy, which is proven by measured violations of Clauser-Horne-Shimony-Holt and Mermin contextuality inequalities. The new capability to provide entangled neutrons can be used for the future studies of entanglement in matter, as suggested by the authors, for example, for the studies of unconventional superconductors.

These capabilities of the entangled neutrons in providing unique probes for the future studies

grants the publication of this paper in Nature Communications journal. I also think the paper is written in a clear and concise way meeting the requirements of the journal.

My only suggestion will be to add a short discussion on where such experiments with entangled neutrons can be conducted in the future and what are the limitations of this method (e.g. materials which can be investigated – should be transparent enough to neutrons, not depolarizing the beam, etc.). Since the reported experiment was conducted at a spallation neutron source, where Time of Flight method was used for the measurement of neutron energy by detectors, can such experiments, where unique neutron probes are generated, be conducted at the continuous sources (e.g. reactor based) and what will be requirements on the energy resolution to be provided at these beamlines.

Some discussion on the practicality of such future experiments will be very useful, I think. For example, what will be the expected intensity of entangled neutron probes, which can be used for the novel studies of matter?

One of the limitations of this method is related to the wavelength range of neutron polarizers, for example, which are not very effective at short wavelengths.

It was also not clear to me from reading the manuscript how the limits of the entanglement lengths (nanometers to micrometers) and energy differences (from peV to neV) were derived by the authors. Is that related to the neutron wavelengths, which can be used in such experiments? The limits are due to the neutron conditioning devices (e.g. polarizers, spin flippers) or by the spectrum/intensity of neutron sources?

Line 68: Value of contextual witness "S" is used here without explanation on what it is. It is only defined later in the text in equation (1).

Response to the Referees

We thank each of the referees for their constructive and perceptive comments. Below we provide detailed responses to each of their suggestions and observations.

Reviewer #1 (Remarks to the Author):

I suggest accepting the manuscript with a few minor edits:

1. reduce the unnecessary use of "entanglement" throughout the article. For example in abstract the phrase "entanglement length" is used, I do not think that that is a proper term.

We have deleted a few instances of the word entanglement. As far as "entanglement length" is concerned we find no better wording that encapsulates its physical meaning. We have realized the neutron entangled probe to explore degrees of freedom of matter that are entangled at microscopic lengths. This is the rationale behind the concept of "entanglement length". Nonetheless, we have replaced the instance of "entanglement length" in the abstract by "spatial separation of trajectories" to avoid introducing an undefined term in the abstract.

2. incorporate a more extensive discussion on: single particle entanglement (correlations between the degrees of freedom) vs the entanglement between different particles. Just for clarity.

We have included a sentence highlighting the distinction between mode and particle entanglement.

3 The experimental setup is similar to <https://www.pnas.org/content/116/41/20328> (which also deals with entanglement between different degrees of freedom, spin and OAM) they should appropriately cite the work.

As suggested by the referee, we have cited <https://www.pnas.org/content/116/41/20328> in the new version. We agree that there are similarities between this PNAS work and ours. It will be interesting in future to simultaneously entangle multiple degrees of freedom including spin, path, energy and OAM.

Reviewer #2 (Remarks to the Author):

From the point of view of neutron interferometry, this paper is important for several reasons. First, the work shows both the high sensitivity and high precision of neutron interferometry as performed on the ISIS LARMOR spin-echo SANS beamline.

As best the reviewer can tell, the LARMOR beamline was used without modification. The sample consisted of a phase prism with an adjustable vertex angle. This means the experiment

pioneered herein is readily duplicated elsewhere, provided beamtime at the world's highly oversubscribed beamlines can be acquired. The publication of this work will certainly improve the odds for successful beamtime proposals.

The referee is correct that we made only slight modifications to the Larmor beamline (we changed the polarization analyzer and the neutron detector from those usually used on Larmor and we operated the RF flippers in resonant mode rather than adiabatic mode). All of these minor changes are described in the Methods section so we have made no further modifications. We agree that the experiment can be repeated elsewhere and we are currently setting up a test at the High Flux Isotope Reactor at Oak Ridge National Laboratory.

The work ends with the statement of potentially widespread applications such as “gravity”. This statement is not backed up with a reference. This reviewer recalls two presentations on a strategy for the use of grating-based interferometry to measure the gravitational constant with 1000-times better accuracy than hitherto possible. The grating-based method is classical; one wonders about the measurement precision with this entanglement method. In summary, the statement of highly interesting applications is strongly supported.

The precision for measurements of the gravitational constant using the entanglement method does indeed need to be thought through in detail and we have not attempted that in our paper. Rather, we have added a sentence on the relationship between precision measurement and quantum metrology which exploits entanglement. In addition, we have added appropriate references to this point, as requested by the referee (reference 20)

This will be the third neutron interferometry quantum entanglement publication in Nature: Hasegawa in 2003 and Clark in 2015. The other two publications have been extremely useful to the community. This work should have similar or stronger impact.

Obviously, we share the referee's opinion and we are working to make entangled neutron scattering a reality

Reviewer #3 (Remarks to the Author):

The manuscript describes a novel experimental technique to entangle neutron beam in spin, trajectory and energy, which is proven by measured violations of Clauser-Horne-Shimony-Holt and Mermin contextuality inequalities. The new capability to provide entangled neutrons can be used for the future studies of entanglement in matter, as suggested by the authors, for example, for the studies of unconventional superconductors.

These capabilities of the entangled neutrons in providing unique probes for the future studies grants the publication of this paper in Nature Communications journal. I also think the paper is written in a clear and concise way meeting the requirements of the journal.

My only suggestion will be to add a short discussion on where such experiments with entangled neutrons can be conducted in the future and what are the limitations of this method (e.g. materials which can be investigated – should be transparent enough to neutrons, not depolarizing the beam, etc.). Since the reported experiment was conducted at a spallation neutron source, where Time of Flight method was used for the measurement of neutron energy by detectors, can such experiments, where unique neutron probes are generated, be conducted at the continuous sources (e.g. reactor based) and what will be requirements on the energy resolution to be provided at these beamlines.

Experiments using a continuous neutron source (the HFIR reactor at Oak Ridge National Lab) are being set up at the moment so we have added a sentence to this effect. Although the experiment we described was done using the TOF method, there was nothing about the experiment that could not have been done with a continuous neutron source. The neutron wavelength resolution that we used, for example, could have been achieved easily at such a source.

Some discussion on the practicality of such future experiments will be very useful, I think. For example, what will be the expected intensity of entangled neutron probes, which can be used for the novel studies of matter?

We expect that experiments with entangled neutrons will be feasible using neutron intensities typical for other uses of polarized neutrons and we have added a statement to this effect.

One of the limitations of this method is related to the wavelength range of neutron polarizers, for example, which are not very effective at short wavelengths.

It is true that neutron polarizers are less efficient at short neutron wavelengths. However, for neutron wavelengths used for most neutron scattering investigations of condensed matter (usually between 0.1 nm and 1 nm), reasonably efficient polarizers already exist. We did not modify the text to make this point since we believe that it is already well known to the neutron scattering community.

It was also not clear to me from reading the manuscript how the limits of the entanglement lengths (nanometers to micrometers) and energy differences (from peV to neV) were derived by the authors. Is that related to the neutron wavelengths, which can be used in such experiments? The limits are due to the neutron conditioning devices (e.g. polarizers, spin flippers) or by the spectrum/intensity of neutron sources?

The referee is correct that we did not explain these limits properly. We have now included some text which we hope answers this observation. In both cases the limits are imposed by the experimental apparatus and could potentially be improved.

Line 68: Value of contextual witness “S” is used here without explanation on what it is. It is only defined later in the text in equation (1).

We agree. We have added a phrase where the witnesses are first mentioned that points the reader to the definition of the witnesses.

Reviewers' Comments:

Reviewer #1:

Remarks to the Author:

The manuscript can be published as is

Reviewer #2:

Remarks to the Author:

This is an excellent manuscript describing excellent work. One minor typo: On page 3, para 3 lines 3 and 11. On line 3, "GHZ" is used without the definition that then follows on line 11.

Publication is highly recommended.

Best wishes

les

Reviewer #3:

Remarks to the Author:

Thanks to the authors for the revised manuscript. All my questions and suggestions were addressed, except for one, related to where such experiments can be conducted in the future.

The authors describe that experiments at a continuous neutron source are planned at the present time. That statement indirectly explains that such experiments can be conducted at a reactor-based source. I would like to suggest to the authors instead of explicitly stating it is possible at a reactor add some direct description on what should be the characteristics of the beam to be used in such experiments: requirements on the energy resolution dE/E , beam divergence, etc. That will allow estimation on whether such experiments can be conducted at a specific facility, rather than stating that it is being planned at HiFIR. Energy resolution, for example, at a continuous source is achieved by either disk choppers, velocity selectors or crystal monochromators, each reducing the flux. Thus requirements on the monochromaticity are important for the estimation of how high the flux will be available at a particular beamline.

Response to referee's comments:

Reviewer #1 (Remarks to the Author):

The manuscript can be published as is

No response required

Reviewer #2 (Remarks to the Author):

This is an excellent manuscript describing excellent work. One minor typo: On page 3, para 3 lines 3 and 11. On line 3, "GHZ" is used without the definition that then follows on line 11.

Publication is highly recommended.

The error has been corrected

Reviewer #3 (Remarks to the Author):

Thanks to the authors for the revised manuscript. All my questions and suggestions were addressed, except for one, related to where such experiments can be conducted in the future.

The authors describe that experiments at a continuous neutron source are planned at the present time. That statement indirectly explains that such experiments can be conducted at a reactor-based source. I would like to suggest to the authors instead of explicitly stating it is possible at a reactor add some direct description on what should be the characteristics of the beam to be used in such experiments: requirements on the energy resolution dE/E , beam divergence, etc. That will allow estimation on whether such experiments can be conducted at a specific facility, rather than stating that it is being planned at HiFIR. Energy resolution, for example, at a continuous source is achieved by either disk choppers, velocity selectors or crystal monochromators, each reducing the flux. Thus requirements on the monochromaticity are important for the estimation of how high the flux will be available at a particular beamline.

We have included a section entitled Pulsed versus continuous neutrons sources on page 5 of the revised manuscript to address the specific issue raised by this referee.